# Enhanced On-State Current and Stability in Heterojunction ITO/ZnO Transistors: A Mechanistic Analysis

**DOI:** 10.3390/nano15030248

**Published:** 2025-02-06

**Authors:** Dengqin Xu, Tingchen Yi, Junchen Dong, Lifeng Liu, Dedong Han, Xing Zhang

**Affiliations:** 1School of Integrated Circuits, Beijing Advanced Innovation Center for Integrated Circuits, Peking University, Beijing 100871, China; 2301111804@pku.edu.cn (D.X.); 2401213287@stu.pku.edu.cn (T.Y.); zhx@pku.edu.cn (X.Z.); 2School of Information & Communication Engineering, Beijing Information Science and Technology University, Beijing 100101, China; jcdong@bistu.edu.cn; 3Beijing Superstring Academy of Memory Technology, Beijing 100176, China; 4School of Software and Microelectronics, Peking University Shenzhen Graduate School, Shenzhen 518055, China

**Keywords:** heterojunction transistor, enhanced on-state current, enhanced stability

## Abstract

The growing demand for high-performance oxide transistors in advanced integrated circuits (ICs) underscores the need for innovative device structures, with heterojunctions emerging as a promising approach. This study presents high-performance ITO/ZnO transistors, which outperform individual ITO or ZnO transistors by achieving an on-state current of 19.2 μA/μm at a drain voltage of 1 V and exhibiting a minimal threshold voltage shift of −0.16 V under negative bias illumination stress. Band structure analysis reveals that the differences in the conduction band minimum and Fermi level between the ZnO and ITO films lead to the formation of a potential well at the ITO/ZnO interface. Furthermore, the increase in the on-state current is attributed to electron confinement at the ITO/ZnO interface, while the enhanced NBIS stability is ascribed to both the band structure and ZnO passivation. These findings make significant contributions to both optimizing the performance and analyzing the mechanisms of oxide devices, highlighting the potential of high-performance ITO/ZnO transistors in 3D integrated circuits, advanced memory devices, and back-end-of-line (BEOL) processes.

## 1. Introduction

Recently, oxide transistors have been successfully commercialized in the field of display, promoting the development of novel applications such as foldable smartphones, ultra-high-definition displays, and smart watches [1,2,3]. Their unique advantages, including high uniformity, low-temperature processing, and good compatibility with Si processes, have also attracted widespread attention in the field of integrated circuits (ICs). So far, novel IC technologies based on oxide transistors have been demonstrated, such as capacitor-less dynamic random access memory (DRAM) [4,5], back-end-of-line (BEOL) process [6], monolithic three-dimensional (M3D) integration [7,8], neuromorphic transistors [9], and ferroelectric transistors [10]. However, oxide transistors, especially those based on InGaZnO (IGZO), are limited by a low field-effect mobility (µ_FE_) below 30 cm^2^/Vs [11,12,13]. A low µ_FE_ results in an insufficient on-state current, hindering their practical applications in ICs. Additionally, their poor stability under bias and illumination, particularly in terms of the negative bias illumination stress (NBIS) stability, limits the long-term operational performance of oxide transistors. Therefore, it is crucial to improve the mobility and stability of oxide transistors through various optimization approaches.

There are several methods for enhancing the on-state current of oxide transistors. One approach involves using high-mobility oxide active layer materials, such as indium-based oxide transistors, especially InSnO (ITO) transistors [14,15,16]. In addition, fabricating a heterojunction active layer, specifically a bilayer structure, can significantly improve the drive current of oxide transistors [17,18,19,20]. Examples include In_2_O_3_/ZnO and IZO/IGZO heterojunction transistors [21,22]. Compared to single-layer active layer oxide transistors, the µ_FE_ of heterojunction transistors shows a substantial increase. Their widely recognized mechanism of action involves electron confinement formed at the heterojunction interface, which leads to the formation of two-dimensional electron gas and, together with the current path at the gate dielectric/active layer interface, facilitates multi-path current conduction [23,24]. For heterojunction transistors, a common design strategy is to use high-mobility materials for the gate dielectric interface and low-mobility materials for the source/drain (S/D) contacts [18]. The high electron density in the front-channel layer results in a high on-state current, while the low electron density in the back-channel layer ensures improved bias stability. However, comprehensive investigations of the heterojunction band structure involving Fermi level (E_F_), valence band maximum (VBM), and conduction band minimum (CBM) are rare. The mechanism for the enhanced bias and illumination stability is not yet fully understood either [25].

Therefore, in this study, we investigated the fundamental mechanisms underlying the performance enhancement in ITO/ZnO transistors. The ITO/ZnO transistors exhibits a significantly improved electrical performance compared to ITO or ZnO transistors, with a high on-state current of 19.2 μA/μm at 1 V, a μ_FE_ of 84.4 cm^2^/Vs at 0.1 V, a subthreshold swing (SS) of 85.7 mV/decade, and an on/off ratio exceeding 10^9^ at a channel length of 10 μm. A comprehensive band structure analysis reveals a 0.33 eV difference in the conduction band minimum (E_CBM_) and a 0.09 eV difference in the E_F_ between ITO and ZnO, resulting in the formation of a potential well at the interface. Electron confinement at the ITO/ZnO interface is the primary factor contributing to the enhanced on-state current. Moreover, in the NBIS test, the ITO/ZnO transistor exhibits the best stability, which can be attributed to both the band structure and ZnO passivation. These findings highlight the advantages of the heterojunction structure in enhancing on-state current and NBIS stability, and deepen our understanding of the mechanisms behind heterojunction transistors, paving the way for their application in advanced ICs.

## 2. Materials and Methods

Figure 1 illustrates the structure and flow of the fabrication process for the heterojunction ITO/ZnO transistors. The gate electrode is an ITO film deposited by radio frequency (RF) sputtering, with an Ar/O_2_ flux ratio of 100/0, while the dielectric is a 9 nm HfO_2_ film formed by atomic layer deposition (ALD). The active layer consists of a 5/12 nm ITO/ZnO bilayer film, where the front ITO layer is deposited by RF sputtering under a pressure of 1 Pa at room temperature, and the back ZnO layer is deposited by ALD with a purging time of 25 s at 120 °C. A Ti/Au bi-layer film, deposited by e-beam evaporation, serves as the S/D electrode. The maximum process temperature does not exceed 120 °C. Device patterning is achieved using conventional photolithography or e-beam lithography, with structure separation performed through a lift-off process. The channel length was determined by the gap length between the S/D electrodes. Additionally, the threshold voltage (Vth) was defined as the gate voltage (V_G_) when the drain current (I_D_) reaches 10 × W/L pA. The electrical performance was evaluated using a semiconductor device analyzer (Agilent B1500A, Keysight Technologies, Santa Rosa, CA, USA). An Atomic Force Microscope (AFM, Bruker Dimension Icon, Bruker Corporation, Billerica, MA, USA) was used to characterize the surface roughness of the films. Ultraviolet Photoelectron Spectroscopy (UPS, Thermo escalab 250XI, Thermo Fisher Scientific, Waltham, MA, USA) and Ultraviolet–Visible Diffuse Reflectance Spectroscopy (UV-Vis DRS, hitachi UH4150, Hitachi High-Tech Corporation, Tokyo, Japan) were employed to investigate the band structure of the transistors. X-ray photoelectron spectroscopy (XPS, Thermo Scientific K-Alpha, Thermo Fisher Scientific, Waltham, MA, USA) was performed to analyze the chemical states of the ITO and ZnO films.

## 3. Results and Discussion

An AFM was employed to characterize the surface roughness of thin films, with a scan area of 4 μm^2^ and a scan resolution of 256 × 256 pixels, as shown in Figure 2a,b. The Root Mean Square Roughness (Rq) for the ITO film was 0.56 nm, while that for the ZnO film was 0.87 nm. Smooth surfaces enhance the electrical performance of the transistors by reducing surface scattering and minimizing surface trap density. In addition, a smooth surface improves the interface quality, ensuring the formation of the heterojunction structure.

The ITO and ZnO transistors were used as the control group. Transfer curves of ITO/ZnO, ITO and ZnO transistors at a channel length of 10 μm are shown in Figure 3a–c. The ITO/ZnO transistors show the best performance, reaching 19.2 μA/μm at a drain voltage (V_D_) of 1 V. The on-state current of ITO/ZnO transistors is significantly higher than that of the ITO transistors (11.9 μA/μm) and the ZnO transistors (1.06 μA/μm). Furthermore, both ITO/ZnO and ITO transistors demonstrate very low leakage currents (below 0.1 pA) and an on/off ratio greater than 10^9^ at a V_D_ of 1 V. The high on/off ratio indicates that these transistors exhibit excellent performance in terms of both leakage current and subthreshold behavior. In addition, the μ_FE_ of ITO/ZnO transistors is 84.4 cm^2^/Vs. The ITO/ZnO transistors exhibit outstanding performances in both on-state and off-state conditions, which make them competitive with previously reported In-based heterojunction transistors, as summarized in Table 1 [26,27,28,29].

Moreover, the Vth values of the ITO/ZnO, ITO, and ZnO transistors are −1.80 V, −1.48 V, and 0.51 V, respectively. A negative Vth shift (∆Vth) is observed in the ITO/ZnO transistors, which correlates with the on-state current. These results imply that the ITO film, which is closer to the dielectric, is the primary determinant of the high on-state current of the transistors, while the ZnO film mainly serves as a current-boosting layer. Therefore, a comparison between the ITO/ZnO and ITO transistors is emphasized in the subsequent discussion to highlight the advantages of the heterojunction structure.

Additionally, the SS of the transistors was investigated and extracted using Formula (1) [22]. The SS values for the ITO/ZnO, ITO, and ZnO transistors are 85.7 mV/decade, 90.2 mV/decade, and 76.7 mV/decade, respectively. The steep SS indicates the excellent gate control provided by the high-k HfO_2_ dielectric. Notably, the ZnO transistors demonstrate the best performance in terms of SS, with the ITO/ZnO and ITO transistors showing slightly inferior results. This can be attributed to the ALD growth of ZnO, which guarantees high-quality films and induces less damage to the interface compared to the RF sputtering process used for ITO/ZnO and ITO.SS = dV_G_/(dlog10I_D_) (1)

The parameters of these transistors at a channel length of 10 μm are summarized in Table 2.

To gain deeper insights into the mechanisms behind the superior electrical performance of the ITO/ZnO transistors over ITO or ZnO transistors, UPS and UV-Vis DRS analyses were conducted to reveal the energy level alignment of ITO and ZnO films. The UV-Vis DRS results are shown in Figure 4a,b. The band gaps (E_g_) of ITO and ZnO were extracted using the Tauc plot method described in Formula (2), considering that both ITO and ZnO are direct bandgap semiconductors.αhv = (hν − E_g_)^(1/2)(2)

Here, α represents the absorption coefficient, h is the Planck’s constant, and ν refers to the incident photon frequency. The E_g_ values were obtained by extrapolating the linear fit of the plot to its intersection with the X-axis. The E_g_ values obtained for ITO and ZnO are 3.33 eV and 3.23 eV, respectively. Additionally, the work functions of ITO and ZnO were measured by UPS, as shown in Figure 4c,d, and calculated using Formula (3). The photon energy (hν) of the helium ion emission line (He I) is 21.22 eV. The work functions (ϕ) of ITO and ZnO were calculated to be 4.22 eV and 4.41 eV, respectively.ϕ = hν − E_cutoff_
(3)

The binding energy of the valence band maximum (E_VBM_) calculated from E_F_ was also measured by UPS, as shown in Figure 4e,f, with values of 2.98 eV and 3.02 eV for ITO and ZnO, respectively.

Based on the E_g_, work function, and the E_f_-E_VBM_, the E_CBM_ energy levels of ITO and ZnO can be determined. A schematic diagram of the heterojunction band structure is illustrated in Figure 5a. The E_F_ of ITO is 0.09 V higher than that of the ZnO, while the calculated E_CBM_ of ZnO is 0.33 eV lower than that of ITO. Upon the formation of heterojunction structure, the electrons from the ITO layer migrate to the ZnO layer to achieve E_F_ alignment across the interface. This electron transfer causes a downward bending of ZnO’s E_CBM_ and an upward bending of ITO’s E_CBM_, creating a potential well at the ITO/ZnO interface. Compared to the single ITO transistor in Figure 5b, this potential well introduces an extra current path in the heterojunction transistor. The corresponding electron transmission paths of ITO/ZnO and ITO transistors are depicted in Figure 5c,d. Considering the excellent electrical performance of the ITO/ZnO transistors together with the UPS and UV-Vis results, we suggest that the conduction band offset at the hetero-interface creates a potential barrier that confines electrons within the interface region. Furthermore, the multiple conductive pathways at the interface facilitate parallel conduction, which increases the charge transport efficiency, thereby contributing to the superior electrical performance of the ITO/ZnO transistors compared to the single ITO or ZnO transistors.

In addition to electrical performance, stability under bias and illumination is also crucial for the practical application of transistors. NBIS is a widely investigated issue in oxide transistors [30,31]. Therefore, in this study, the NBIS behaviors of these transistors were also examined, as shown in Figure 6a–c. The electric field used in the NBIS tests was −10^6^ V/cm, and the illumination intensity was a mixed white light at 10,000 Lux. The ∆Vth values under different bias stress durations are presented in Figure 6d. The Vth of the ITO transistors drifts negatively, with a maximum shift of −0.36 V after 1000 s. The ZnO transistors also demonstrate a negative ∆Vth of −0.2 V after 1000 s. In contrast, the Vth of ITO/ZnO transistors exhibits only a minimal negative drift of −0.16 V after 1000 s. This indicates that ITO/ZnO transistors have enhanced NBIS stability compared to single-layer transistors. The NBIS of ITO/ZnO transistors is competitive with previous studies [32,33,34]. To better understand relationship between ∆Vth and stress time, the curves in Figure 6d were fitted using the stretched exponential formula, as shown in Formula (4) [35,36].∆Vth = ∆V_max_ {1 − exp [−(t/τ)^β]}(4)

Here, the τ refers to the characteristic time constant, while β represents the stretched exponential. The stretched exponential behavior is related to the charge trapping mechanism [37]. The β values for all three types of oxide transistors were calculated to be approximately 1, indicating that the time dependence of ∆Vth is dominated by a single time constant τ and that the ∆Vth process is primarily controlled by a single dynamic mechanism. Considering the negative ∆Vth, we propose that this mechanism is mainly driven by the capture of photo-generated electrons by ionized oxygen vacancies (V_O_^2+^). The photo-generated electrons excite the neutral oxygen vacancy (V_O_), causing them to transition to V_O_^2+^ and release two free electrons, leading to the negative ∆Vth observed.

The τ values of the ITO/ZnO, ZnO, and ITO are 194, 186, and 145 s, respectively. The τ is closely related to the V_O_ concentration. A higher concentration of V_O_ promotes the transition of V_O_ to V_O_^2+^, resulting in a shorter τ. To investigate the chemical states of the O element in ITO and ZnO films, an XPS analysis was conducted. The XPS spectra were calibrated using the C 1s peaks at 284.8 eV. The XPS survey spectra for ITO and ZnO films are shown in Figure 7a,b, with no significant impurity peaks observed. The O 1s characteristic peaks for both ITO and ZnO films were split into two characteristic peaks, as shown in Figure 7c,d. For ITO, the peak located at 529.8 eV corresponds to lattice oxygen (O-M), and the peak at 531.44 eV is attributed to Vo^2+^ [38]. The V_O_ concentration is estimated based on the XPS O 1s peak area ratio. The concentration of V_O_ in ITO is approximately 50.6%. For ZnO, similarly, the characteristic peaks observed at 530.33 eV and 531.94 eV are attributed to the O-M bond and V_O_^2+^, respectively [39]. The concentration of V_O_ in ZnO is 32.4%. The significantly lower V_O_ concentration in ZnO compared to ITO explains the smaller τ and better NBIS performance of the ZnO transistors.

The improved NBIS performance in ITO/ZnO transistors can be explained by the band structure of the ITO/ZnO heterojunction. As discussed earlier, electrons accumulate at the interface between the ITO and ZnO films, which helps to suppress trap states and the formation of V_O_^2+^ at the interface. The electrons in the potential well will combine with the V_O_^2+^ to re-form V_O_. Additionally, the improvement in NBIS behavior is also attributed to the ZnO film acting as a passivation layer, isolating the ITO from the air due to its lower V_O_ concentration compared to ITO. Considering the ultra-thin nature of the ITO layer, optimizing the interface becomes even more crucial for improving NBIS stability. Furthermore, the band offset at the interface of ITO and ZnO films forms a barrier that prevents photogenerated electrons and holes in ZnO from entering the ITO, thereby contributing to improved NBIS stability [40]. In brief, the heterojunction structure effectively enhances both the electrical performance and NBIS stability of the transistors.

## 4. Conclusions

In this work, high-performance ITO/ZnO heterojunction transistors were successfully fabricated. These ITO/ZnO transistors exhibit a μ_FE_ of 84.4 cm^2^/Vs at a V_D_ of 0.1 V, an on-state current reaching 19.2 μA/μm at a V_D_ of 1 V, a SS of 85.7 mV/decade, and an on/off ratio over 10^9^. Notably, they also demonstrate enhanced NBIS stability, with a ∆Vth of only −0.16 V after a 1000 s test. UPS and UV-Vis DRS analyses reveal a 0.33 eV ∆E_CBM_ and a 0.09 V ∆E_F_ between the ITO and ZnO films. This results in the formation of a potential well at the interface, which is proposed to enhance the on-state current through electron confinement. Additionally, analysis using the stretched-exponential formula indicates that the capture of photo-generated electrons plays a dominant role in NBIS stability. Along with the band structure and XPS analysis, we believe that the improvement in NBIS stability stems from two factors: the potential well at the interface suppressing the V_O_ ionization, and the ZnO back-channel layer with a low V_O_ concentration acting as a passivation layer. The competitive electrical characteristics of our low-temperature-processed ITO/ZnO transistors highlight their potential for applications in the BEOL process, 3D ICs, and novel memory technologies. On top of that, our analysis of on-state current and NBIS stability enhancement in ITO/ZnO is expected to further enrich our understanding of the underlying mechanism of heterojunction transistors.

## Figures and Tables

**Figure 1 nanomaterials-15-00248-f001:**
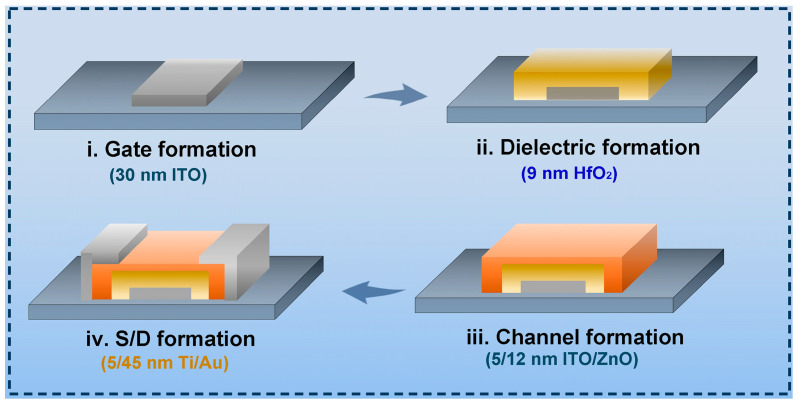
Device structure and process flow of ITO/ZnO transistors.

**Figure 2 nanomaterials-15-00248-f002:**
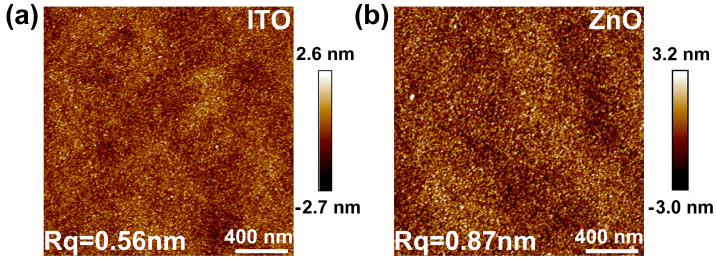
(**a**,**b**) AFM images of ITO and ZnO films.

**Figure 3 nanomaterials-15-00248-f003:**
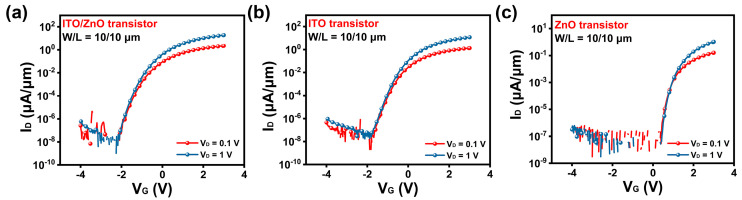
(**a**–**c**) Transfer curves of ITO/ZnO, ITO, and ZnO transistors at a channel length of 10 μm.

**Figure 4 nanomaterials-15-00248-f004:**
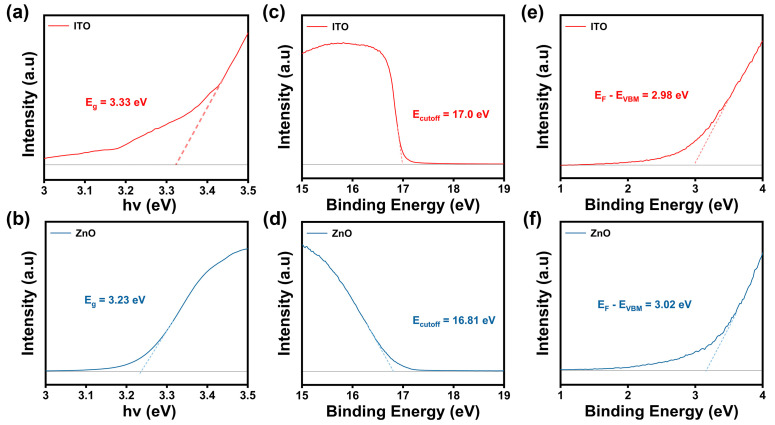
(**a**,**b**) E_g_ of ITO film and ZnO film extracted using Tauc method; (**c**,**d**) cutoff energy (E_cutoff_) of ITO film and ZnO film measured by UPS analysis; (**e**,**f**) binding energy of E_VBM_ measured from E_F_ of ITO film and ZnO film.

**Figure 5 nanomaterials-15-00248-f005:**
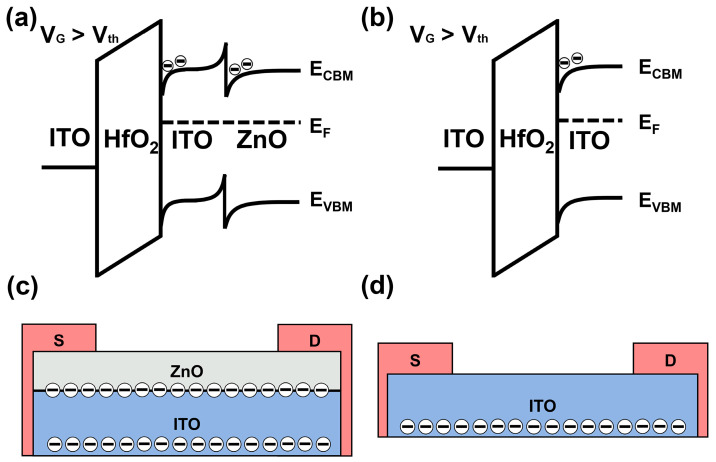
(**a**) Schematic energy band diagrams of the ITO/ZnO transistors at V_G_ > Vth for the electron confinement formation; (**b**) schematic energy band diagrams of the ITO transistors at V_G_ > Vth; (**c**,**d**) transmission paths of the electrons in ITO/ZnO and ITO transistors.

**Figure 6 nanomaterials-15-00248-f006:**
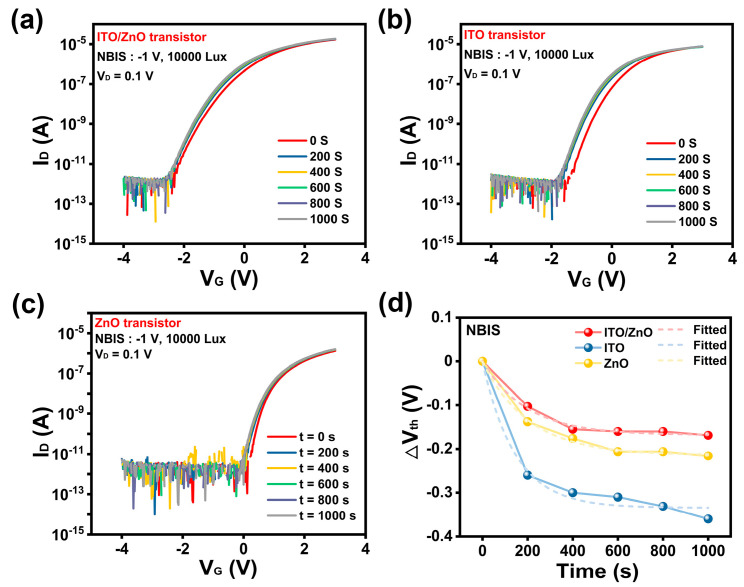
(**a**–**c**) Transfer curves of ITO/ZnO, ITO, and ZnO transistors under NBIS, respectively; (**d**) comparison of the extracted ∆Vth of three types of transistors in the NBIS test.

**Figure 7 nanomaterials-15-00248-f007:**
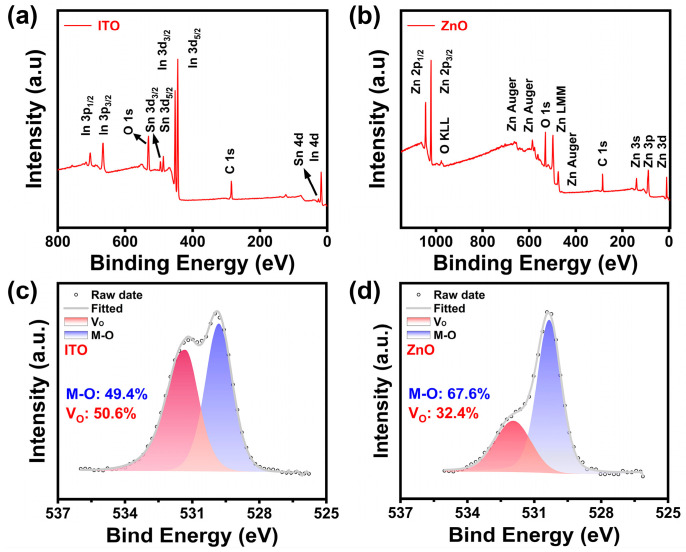
(**a**) XPS survey spectrum of the ITO film; (**b**) XPS survey spectrum of the ZnO film. (**c**) O 1s XPS spectrum of the ITO film; (**d**) O 1s XPS spectrum of the ZnO film.

**Table 1 nanomaterials-15-00248-t001:** Benchmarking of ITO/ZnO transistor.

Sample	μ_FE_ cm^2^/Vs	Leakage Current pA	On/Off
ITO/ZnO (this work)	84.4	0.1	over 10^9^
ITO/ZTO [26]	~50	1	over 10^8^
In_2_O_3_/ZnO [27]	~50	10–100	~10^8^
In_2_O_3_/IZO [28]	37.9	0.1	~10^9^
In_2_O_3_/ZnO [29]	~50	10	~10^7^

**Table 2 nanomaterials-15-00248-t002:** Parameter of ITO/ZnO and ITO transistors at a channel length of 10 μm and a V_D_ of 1 V.

Sample	On-State Current μA/μm	Vth V	SS mV/Decade
ITO/ZnO	19.2	−1.80	85.7
ITO	11.9	−1.48	90.2
ZnO	1.06	0.51	76.7

## Data Availability

The data presented in this study are available on request from the corresponding author.

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
