# Peer review of "Enhanced On-State Current and Stability in Heterojunction ITO/ZnO Transistors: A Mechanistic Analysis"

_nanomaterials, 2025, doi:10.3390/nano15030248_

Round 1

Reviewer 1 Report

Comments and Suggestions for Authors

The manuscript describes an ITO/ZnO heterojunction transistor at high performance. The fabrication steps of transistor are described. The electrical mechanism is attributed to the electron confinement at the interface enhancing on-state current. The results are interesting but the work is poorly written and needs revisions and further adds. Some Major Revisions are suggested:

a. XPS investigations have been done, but other material characterizations are necessary such as SEM, XRD and Raman spectroscopy. These images and spectra should be provided to appreciate any potential correlation between electrical properties and material microstructure.

b. what is about on process repeatability of ITO and ZnO material for transistor ? Please, comment on this aspect.

c. comparison of ITO/ZnO transistor performance with literature data should be provided by a table. Please, comment on this.

d. other references should be added in bibliography.

e. figures need to be higher size to appreciate details at better resolution.

Comments on the Quality of English Language

English could be improved.

Reviewer 2 Report

Comments and Suggestions for Authors

The paper deals with the ITO/ZnO transistors promising for many applications such as smart watches, smartphones etc. This is hot and important topic. The paper is focused on the ITO/ZnO heterojunction stability. The obtained results show that this heterojunction transistors  demonstrate high performance and on/off ratio over 10**9. The potential well at the interface is predicted, based on the analysis of UPS and UV-Vis DRS data. The reasons for the stability are discussed. These are new and original results. The cited references are appropriate. The paper could be published after minor revision.

1. Estimate error bars in the band gaps for ITO and ZnO (fig.3) which differ only by 0.1 eV.

2.  Explain how defect concentrations Vo were calculated, is it with respect to what? Total number of O atoms? how does it depend on the experimental conditions? Does Vo mean neutral vacancies (2 trapped electrons)?

3.  The English needs some polishing, e.g. line 10 above Conclusions: Since...

4. In references, give only year, not publication month (e.g. Dec 4, 2024), add year in ref. 14 

5. The conclusion on the potential well at the interface could be checked with ab initio calculations

Round 2

Reviewer 1 Report

Comments and Suggestions for Authors

The revised manuscript has been enhanced. The questions risen by this referee have been fairly answered. The revised manuscript is now suitable for publication.